# *Panax quinquefolius* (North American Ginseng) Polysaccharides as Immunomodulators: Current Research Status and Future Directions

**DOI:** 10.3390/molecules25245854

**Published:** 2020-12-11

**Authors:** Rajarshi Ghosh, Daniel L. Bryant, Anthony L. Farone

**Affiliations:** 1Department of Biochemistry, University of Missouri, Columbia, MO 65211, USA; rgf8b@missouri.edu; 2Department of Biology, Middle Tennessee State University, Murfreesboro, TN 37132, USA; dlb6n@mtmail.mtsu.edu; 3Tennessee Center for Botanical Medicine Research, Middle Tennessee State University, Murfreesboro, TN 37132, USA

**Keywords:** ginseng, plant polysaccharides, immunostimulant, mechanism of action

## Abstract

*Panax quinquefolius* (North American ginseng, NAG) is a popular medicinal plant used widely in traditional medicine. NAG products are currently available in various forms such as roots, extracts, nutraceuticals, dietary supplements, energy drinks, etc. NAG polysaccharides are recognized as one of the major bioactive ingredients. However, most NAG reviews are focused on ginsenosides with little information on polysaccharides. NAG polysaccharides have demonstrated a therapeutic activity in numerous studies, in which many of the bioactivities involve regulation of the immune response. The purpose of this review is to summarize the structural features and the immunomodulatory properties of crude, partially purified, and pure polysaccharides isolated from NAG. Receptors of the innate immune system that potentially bind to NAG polysaccharides and the respective signal transduction pathways initiated by these compounds are discussed. Major challenges, recent innovations, and future directions in NAG polysaccharide research are also summarized.

## 1. Introduction

Medicinal plants have been used successfully for disease prevention and treatment in many cultures worldwide for centuries. This popularity has reached unparalleled levels in western society in recent years, as well. Several United States FDA-approved pharmaceuticals are plant natural products or derivatives [1]. Apart from pharmaceuticals, plants contribute greatly to an ever-expanding nutraceutical and dietary supplement market. The global nutraceutical market is estimated to be valued at over USD 230 billion and is expected to reach USD 336 billion by 2023 [2]. Numerous bioactive plant secondary metabolites have been studied intensively for decades and subsequently commercialized as nutraceuticals. On the other hand, bioactive macromolecules from plants have comparatively received much less attention. Interestingly, over the last two decades, research on plant polysaccharides has revealed their roles in the human cellular function and metabolism. Complex carbohydrates present in common fruits and vegetables, as well as medicinal plants may contribute to the functional properties associated with the particular plant [3]. These high molecular weight compounds have shown promise to be developed into nutraceuticals suitable for the treatment of various diseases and metabolic disorders.

A wide range of plant polysaccharides have demonstrated diverse bioactivities such as immunomodulatory [4], anticancer [5], neuroprotective [6], antioxidant [7], and antimicrobial [8] properties. Several articles have discussed the ability of plant polysaccharides to modulate the immune system without causing any adverse side effects [4,9]. It is widely believed that the majority of the bioactivities caused by polysaccharides are an indirect effect of immunomodulation [9]. The strategy to activate or suppress the immune system by natural or synthetic immunomodulators has become increasingly popular in the treatment of various types of infections and cancers [10]. Plant polysaccharides are natural exogenous immunomodulators that have been shown to be effective nutraceuticals with a broad-spectrum therapeutic activity. Additionally, polysaccharides can play an important role in existing immunotherapies and vaccination practices [11]. The immunostimulatory properties of these compounds may be utilized as adjuvants and wound healing agents. Trends in the current research indicate that the application of polysaccharide-based therapeutics in human nutrition and health has an immense potential.

Ginseng, one of the most famous medicinal herbs worldwide, has been used for thousands of years in traditional medicine. Among the several different types of ginseng, the two most prominent species are *Panax ginseng* (Asian ginseng) and *Panax quinquefolius* (North American ginseng; NAG). Both species are a rich source of bioactive phytochemicals that have been extensively used in various food products, nutraceuticals, and dietary supplements. Along with triterpenoid saponins known as ginsenosides, polysaccharides are recognized as one of the major bioactive ingredients in ginseng [12,13]. Different classes of polysaccharides are highly abundant in ginseng and constitute about 10% (*w*/*w*) of dried ginseng roots [12]. Ginseng polysaccharides have demonstrated an impressive therapeutic potential in several in vitro and in vivo studies [12,14]. Due to the interest in the medicinal properties of ginseng, there have been several reviews written on both Asian and North American ginseng [13,15,16]. However, most ginseng reviews have focused primarily on the bioactivity of small molecules such as ginsenosides [17,18,19]. Comprehensive reviews on ginseng polysaccharides are far fewer compared to the secondary metabolites. Recently, a number of review articles have been published highlighting naturally occurring bioactive polysaccharides in Asian ginseng [6,20,21]. This review will focus on the polysaccharides isolated from different parts of the NAG plant, their immunomodulatory properties and putative mechanisms of action, and novel approaches towards the successful application of NAG polysaccharides as nutraceuticals.

## 2. Current Research Status of NAG Polysaccharides: An Overview

*Panax quinquefolius* is a perennial herb belonging to the Araliaceae family. It is native to eastern North America and widely inhabits several Canadian and US states. The species is also cultivated in China since the 1980s [19]. NAG roots are a rich source of both ginsenosides as well as polysaccharides [12,19]. Wild grown ginseng roots are generally harvested at 5–7 years of growth or later, whereas farm-grown NAG is harvested after 3–4 years of growth. The farm-grown products are generally considered substandard compared to wild-grown ginseng roots [22,23]. The increasing demand for wild ginseng has led to dwindling natural habitats and due to conservation concerns, cultivation of farm-grown ginseng is encouraged. Currently, NAG is widely cultivated in the US states of Wisconsin, Michigan, North Carolina, Ohio, and Tennessee, as well as several Canadian provinces such as Ontario and British Columbia [24]. A number of research institutes dedicated to NAG research have also been established in North America [25,26]. A greater focus on NAG research has led to the isolation and characterization of bioactive phytochemicals including polysaccharides and the subsequent commercialization of NAG nutraceuticals. NAG products are currently available in the form of whole, sliced or powdered roots, extracts and tablets. NAG can also be found in several dietary supplements, health products, energy drinks, as well as shower gels, lotions, and shampoos [19]. One such polysaccharide-based NAG natural health product that has become quite popular is CVT-E002 (commercially sold as COLD-FX). This proprietary poly-furanosyl-pyranosyl polysaccharide-rich extract has been extensively researched for over a decade. Some of the in vitro, in vivo, and clinical trials using CVT-E002 will be reviewed in a later section. There has been several other examples of crude and partially purified polysaccharides that have shown promise in immunomodulatory studies. Additionally, a number of bioactive polysaccharides have been purified from NAG in recent years. The structures of these purified polysaccharides, their immunomodulatory properties, putative mechanisms of action, and potential for development into nutraceuticals are discussed in this review.

Although, the progress of NAG polysaccharide research over the last decade has been promising, there are several challenges that need to be addressed. Batch to batch structural variations (e.g., variation in size, monosaccharide composition, glycosidic linkage, and degree of branching) in most plant polysaccharides including NAG are common. Changes in environmental conditions and stress can often lead to these variations. Inconsistent structural characteristics may lead to poor standardization of nutraceuticals. Since polysaccharides are high molecular weight compounds, there are concerns regarding its bioavailability and eventual efficacy. Another major issue that needs to be addressed in medicinal plant immunomodulation studies is lipopolysaccharide (LPS) contamination in herbal preparations [27,28]. Minute amounts of LPS can lead to false positives in immunomodulatory assays. LPS contamination can lead to misleading claims about polysaccharide-based nutraceuticals, as well as safety issues. Rigorous quality control is often overlooked in the nutraceutical industry due to the lack of uniform regulations worldwide. Addressing these issues will be key towards the success of NAG polysaccharides as safe and effective nutraceuticals in the upcoming years. This review highlights some of the recent innovations in NAG polysaccharide research that can begin to address some of these issues.

## 3. Extraction, Purification, and Characterization of NAG Polysaccharides

Researchers have used NAG aqueous extracts [29], crude polysaccharide extracts [30], partially purified heterogeneous polysaccharide fractions [12], as well as homogeneous polysaccharides [31] to elicit an immune response in vitro and in vivo. A wide range of extraction and purification strategies are currently employed to prepare NAG polysaccharide samples. The most commonly used method to extract water-soluble components in NAG tissues is hot water extraction. Powdered NAG tissues are boiled for several hours two to three times and subsequently filtered and concentrated under reduced pressure. Although this method is the easiest, it has several disadvantages such as low extraction efficiency, long extraction times, and high extraction temperature [20]. Another method to prepare the crude polysaccharide extract devoid of lipophilic compounds and other small molecule impurities is to initially extract powdered NAG tissues with 95% ethanol [14]. Extraction with ethanol likely removes small molecule impurities. The residue is then extracted with hot water, filtered, and further deproteinated using the Sevag reagent (n-butanol: chloroform = 1:4) [32]. Crude polysaccharide is then precipitated from the deproteinated extract using ethanol and dissolved in diH_2_O. Although most polysaccharides can be extracted using this method, some high molecular weight polysaccharides are not easily extracted with hot water. The dilute alkali-water solution is generally used to extract the remaining polysaccharides in the residue after hot water extraction. Yu et al. (2014) used a 0.3 M NaOH solution with 0.3% (*w*/*w*) NaBH_4_ to extract two alkali-soluble polysaccharides from NAG roots [31]. Another method that is being increasingly used to extract polysaccharides is enzyme-assisted extraction. An aqueous or crude polysaccharide extract is mixed with α-amylase and protease and hydrolyzed overnight. The hydrolyzed extract is then dialyzed to obtain a polysaccharide solution devoid of starch and proteins [33]. Other polysaccharide extraction methods include ultrasonic extraction, microwave extraction, and supercritical fluid extraction using CO_2_ [20,34]. Some of these new and improved extraction methods are optimized to improve polysaccharide extraction efficiency, decrease extraction time, and reduce solvent consumption. These optimized extraction methods are crucial towards the successful application of NAG polysaccharides as nutraceuticals.

Most commercially available polysaccharide-rich NAG nutraceuticals are crude extracts. The purification of crude polysaccharide extracts often leads to high costs and lower yields unsuitable for commercial applications. However, due to the complex nature of crude polysaccharide extracts, it is difficult to identify bioactive glycans and determine structure-activity relationships. In order to identify specific glycan moieties, it is essential to purify polysaccharides, elucidate their structural features, and conduct bioactivity studies in a mechanistic manner. In recent years, researchers have used lower cost purification methods such as graded ethanol precipitation, dialysis, and centrifugal filtration with different molecular weight cut-offs to partially purify ginseng polysaccharides [12,20]. However, column chromatography continues to be the preferred method to purify plant polysaccharides. Crude polysaccharides are generally fractionated using anion exchange columns (e.g., DEAE-Sepharose) and eluted using a linear or stepwise gradient of NaCl. The water-eluted neutral fraction and the NaCl-eluted acidic fractions are further purified using size-exclusion chromatography. A number of different size-exclusion resins (e.g., Sephadex, Sephacryl, Superdex etc.) can be used to purify polysaccharides. Sequential anion-exchange and size-exclusion chromatography have been effectively used by several researchers to purify NAG polysaccharides [14,35]. Ideally, purified polysaccharides should be further eluted through an endotoxin removal resin (e.g., Pierce high capacity endotoxin removal resin) prior to the bioactivity screening [14,36,37]. Although this additional step may reduce the polysaccharide yield, it is vital to rule out potential false positives in immunomodulatory assays. The extraction and purification steps are described in Figure 1.

Structure elucidation of plant polysaccharides is challenging due to their heterogeneity and diversity. A combination of biochemical and spectroscopic methods has been used to characterize NAG polysaccharides. A high-performance size exclusion chromatography coupled with a refractive index (HPSEC-RI) detector is the typically used method to determine homogeneity and the molecular weight of purified polysaccharides. Recently, a multi-angle laser light scattering (MALLS) detector has become the preferred choice for molecular weight analysis of polysaccharides due to its increased performance [38,39]. For the monosaccharide composition analysis, polysaccharides are generally hydrolyzed, derivatized (e.g., trimethylsilyl derivatization for GC-MS and 1-phenyl-3-methyl-5-pyrazolone derivatization for HPLC), and analyzed by GC-MS or HPLC [14,36]. The analysis of underivatized monosaccharides is becoming increasingly popular after the advent of high-performance anion exchange chromatography-pulsed amphoteric detection (HPAEC-PAD) [40]. HPLC-coupled to semi-universal detectors such as the evaporative light scattering detector (ELSD) and charged aerosol detector (CAD) can also analyze the monosaccharide composition of certain polysaccharides without derivatization [41,42,43]. Although HPAEC-PAD and HPLC-ELSD/CAD methods have not been used for the analysis of unknown NAG polysaccharides thus far, these techniques will eventually be more routinely used for the analysis of unknown compounds as chromatographic methods are further optimized. Apart from the monosaccharide composition, glycosidic linkages also reveal substantial information about the primary structure of polysaccharides. Samples are permethylated, hydrolyzed, reduced, acetylated, and the resulting partially methylated alditol acetates (PMAA) are analyzed by GC-MS for linkage analysis of polysaccharides [14,37]. However, the complete structure of polysaccharides including the sequence of constituent monosaccharides cannot be obtained by the above-mentioned methods. Anomeric conformations, sequence of monosaccharides, and substitution patterns in polysaccharides are obtained by extensive 1D and 2D NMR experiments [33]. Other techniques that have been commonly used to elucidate structures of NAG polysaccharides include Fourier transform infrared spectroscopy (FT-IR). FT-IR can indicate the presence of uronic acids, as well as the presence of α and β sugar residues in polysaccharides [20,36]. Overall, a combination of complimentary analytical techniques is required to elucidate the structural features of these complex carbohydrates. A comprehensive list of modern analytical techniques for herbal glycomics was reviewed by Li et al. (2013) [44].

A large number of early pharmacological studies conducted with polysaccharide-rich NAG aqueous extracts or crude polysaccharides lacked structural information. Minimum structural information such as sugar content, protein content, and ratio of monosaccharide constituents in crude extracts are important for quality control, as well as reproducibility. Although studies in the past decade generally adhered to a set of minimum reporting standards, there are still several examples that lack the adequate structural information regarding extracts. Out of the several NAG extracts that have been reported, CVT-E002 or COLD-FX is very well known. It is a proprietary extract containing >80% poly-furanosyl-pyranosyl saccharides with no ginsenosides. Monosaccharide constituents have been reported to be Glc > GalA > Ara > Gal > Rha [45]. There has also been significant progress towards the purification and characterization of several polysaccharides from the roots of NAG. One of the earliest studies that attempted to purify and elucidate structures of NAG polysaccharides was conducted by Oshima et al. (1987) [46]. Three polysaccharides (Quinquefolan A–C) were purified by anion exchange and size exclusion chromatography. Following the purification of the quinquefolans, there were reports of a few purified NAG polysaccharides in China in the 1990s, which were unavailable in English. In the last decade, there has been an increase in the number of publications focusing on purified NAG polysaccharides that have been partially or fully characterized. Zhu et al. (2012) isolated a homogenous glucogalactan named PPQ from the roots of NAG [47]. Yu et al. (2014) extracted two polysaccharides namely AEP-1 and AEP-2 using dilute alkali from NAG roots [31]. Due to the lack of glycosyl linkage and NMR data, it is difficult to predict the specific polysaccharide classes of AEP-1 and AEP-2. A neutral polysaccharide named PPQN similar to the glucogalactan reported by Zhu et al. (2012) [47] was isolated from NAG roots by Wang et al. (2015) [48]. In a separate study, Wang et al. (2015) isolated three acidic polysaccharides PPQA2, PPQA4, and PPQA5 from the roots of NAG. Although molecular weights and monosaccharide compositions were reported, complete structural characterizations of PPQN, PPQA2, PPQA4, and PPQA5 by 2D NMR were not carried out. Non-starch polysaccharides were also isolated from NAG roots by enzyme-assisted extraction [33]. One of the polysaccharides GSP was reported to be a pectic polysaccharide with a major homogalacturonan domain and a minor rhamnogalacturonan-I (RG-I) component. Yu et al. (2017) isolated and partially characterized two neutral (WPS-1 and WPS-2) and three acidic (SPS-1, SPS-2, and SPS-3) polysaccharides from NAG roots [35]. Although the study reported molecular weight, monosaccharide composition, ^1^H and ^13^C NMR spectra of the isolated polysaccharides, glycosidic linkage analyses, and 2D NMR experiments were not carried out to completely characterize these compounds. Overall, less than fifteen NAG root polysaccharides have been purified and characterized thus far. This number is much lower compared to the Asian ginseng where close to eighty polysaccharides have been purified and analyzed [20]. Apart from NAG root polysaccharides, there are very few reports of characterized polysaccharides from the aerial parts of the plant. While NAG roots will continue to be heavily researched in the upcoming years, the aerial parts of NAG offer great opportunities for purification and chemical structure elucidation studies. Recently, there has also been an initiative to isolate bioactive polysaccharides from NAG callus and suspension cultures. The plant tissue culture offers a unique platform for the production of standardized natural polysaccharides with reduced endotoxin content [14]. More than five polysaccharides with immunomodulatory properties have been isolated from NAG suspension cultures. Ghosh et al. (2019) purified a neutral polysaccharide fraction AGC1 that was predominantly composed of arabinogalactan type II polysaccharides. The structural features of another acidic polysaccharide fraction (AGC3) from the NAG suspension culture was reported by Ghosh et al. (2020) [37]. The linkage analysis indicated the presence of the RG-I type of pectic polysaccharides in AGC3. The complete list of purified NAG polysaccharides along with their structural features are described in Table 1.

The last decade has seen great progress regarding the isolation and structure elucidation of bioactive NAG polysaccharides. We expect more NAG polysaccharides to be fully characterized in the upcoming years as modern analytical techniques for carbohydrate characterization continue to improve. The complete structures of these polysaccharides will eventually pave the way for structure-activity studies and identification of specific bioactive glycan moieties. The next two sections will describe the immunomodulatory properties of NAG crude polysaccharide extracts, purified NAG polysaccharides, and their putative mechanisms of immunomodulation.

## 4. Immunomodulatory Properties of NAG Polysaccharide Extracts

The initial research demonstrating the immunomodulatory properties of NAG polysaccharides were conducted mostly using crude extracts. Some noteworthy studies have been highlighted in this section. Assinewe et al. (2002), showed that NAG polysaccharide extracts, not the ginsenosides, were responsible for the immunostimulatory properties of the plant [29]. In that study, a NAG crude polysaccharide fraction composed of Glc (85.09%), Gal 97.48%), Ara (5.89%), Fuc (0.09%), Rha (0.79%), and Man (0.41%) significantly stimulated tumor necrosis factor-α (TNF-α) in rat alveolar macrophages in vitro. Lemmon et al. (2012) similarly showed that NAG aqueous and crude polysaccharide extracts significantly induced interleukin-6 (IL-6), IL-1β, TNF-α, and IL-10 production in human peripheral blood mononuclear cells (PBMC) [30]. They concluded that high molecular weight NAG polysaccharides triggered an immunomodulatory response characterized by a net Th_1_ immune response in PBMC. According to the study, the induction of Th_1_ transcriptional profile was likely triggered by mitogen activated protein kinase (MAPK), nuclear factor-κB (NF-κB), and phosphoinositide 3-kinase (PI3K) signaling pathways. Wilson et al. (2013) also demonstrated that the NAG polysaccharide extract induced a global inflammatory response in differentiated preadipocytes in vitro [49]. 3T3-L1 preadipocytes treated with the NAG polysaccharide extract significantly increased the expression of IL-6, TNF-α, and NF-κB. The authors indicated that Toll-like receptor-4 (TLR4) likely plays a major role towards the upregulation of inflammatory gene and protein expressions. Azike et al. (2015) demonstrated the immunomodulatory properties of NAG aqueous extract, crude polysaccharide extract, and partially purified polysaccharide fractions in vitro, ex vivo, and in vivo [12]. The polysaccharides increased nitric oxide (NO) and TNF-α production in RAW 264.7 murine macrophage cells (in vitro) and rat alveolar macrophages (ex vivo) [12]. Plasma NO and TNF-α levels were significantly elevated in adult Sprague-Dawley rats, which were orally administered with NAG polysaccharide extracts [12]. Interestingly, the same extracts suppressed the LPS-induced elevation of the proinflammatory mediators NO and TNF-α [12]. This study indicated that NAG polysaccharides may possess a paradoxical immunostimulatory, as well as immunosuppressive properties. The authors further concluded that the immunomodulatory effects of the extracts were primarily mediated by high molecular weight (>50 kDa) acidic polysaccharides. There is considerable debate regarding the dual stimulatory and anti-inflammatory properties of polysaccharides and further research needs to be conducted to clarify these findings. Overall, the immunomodulatory effects of NAG polysaccharides have been detailed in Table 2. Additionally, a number of studies have shown immunomodulatory properties of crude NAG extracts containing both ginsenosides and polysaccharides [50,51]. Extracts containing both ginsenosides and polysaccharides will not be discussed in this review.

One of the most well-known polysaccharide-based NAG natural health products CVT-E002 or COLD-FX has also demonstrated immunomodulatory properties in several studies (Table 3) conducted over a period of more than 10 years. Wang et al. (2001) showed that CVT-E002 significantly increased murine spleen B lymphocyte proliferation ex vivo and serum immunoglobulin G (IgG) production in vivo [45]. The standardized polysaccharide fraction also stimulated proinflammatory mediators such as IL-1, IL-6, TNF-α, and NO in peritoneal exudate macrophages. In a separate study, Wang et al. (2004) showed that CVT-E002 significantly increased concanavalin A (ConA)-induced murine splenocytic production of IL-2 and interferon-γ (IFN-γ) ex vivo [55]. These results indicated that CVT-E002 can potentially modulate both innate and acquired immune response.

However, in another study conducted by Biondo et al. (2008), CVT-E002 consumption decreased IL-2 and IFN-γ production in ConA or LPS-stimulated lymphocytes isolated from the spleen, mesenteric lymph nodes, and Peyer’s patches of male weanling Sprague–Dawley rats [56]. The TNF-α production did not differ and only IL-1β significantly increased between the experimental and control groups [56]. The reduction in IL-2 and IFN-γ was likely a result of decreased CD3+ and activated T-cells [56]. The authors concluded that the immunomodulatory property of CVT-E002 is not likely a result of splenic Th_1_ and Tc_1_ cytokine production. A number of randomized double-blind, placebo-controlled trials have also been conducted demonstrating the effectiveness of CVT-E002 against upper respiratory tract infections (URI) [57,60,61]. The extract reduced the mean number of colds per person and the severity of common cold-like symptoms in a study involving 323 subjects conducted by Predy et al. (2005) [60]. Subjects given CVT-E002 showed an increased proportion of T-helper cells and natural killer (NK) cells and decreased plasma IgA levels compared to the placebo. Although the exact mechanism of action was unclear, the authors speculated that Th cells and NK cells acted synergistically to reduce the severity and duration of URIs [61]. Another placebo-controlled trial by McElhaney et al. (2004) reported that CVT-E002 was safe, well-tolerated, and potentially effective against acute respiratory illness in older institutionalized adults [57]. McElhaney et al. (2010) also reported that cultured peripheral blood leukocytes (PBL) isolated from healthy athletes treated with CVT-E002 and infected with three strains of influenza virus, released greater TNF-α and IL-2 compared to the control group [58]. The authors speculated that the CVT-E002 supplementation enhanced a cell-mediated immune response of PBLs to influenza viruses and the reported enhanced immune response is likely moderated by cytotoxic T-cells. Overall, these studies offer important insights regarding the immunomodulatory activity of CVT-E002, but the mechanisms of action remain unclear. The immunomodulatory properties of NAG polysaccharides including COLD-FX are depicted in Figure 2.

We expect that NAG crude polysaccharide extracts with immunomodulatory properties will continue to be formulated in the upcoming years. It is true that standardized crude extracts are economically more feasible as nutraceuticals compared to purified compounds. However, purified polysaccharides have a well-defined mechanism of action. The key towards proper and credible formulation and application of these plant polysaccharides as therapeutics will eventually depend on the elucidation of structure-activity relationships.

## 5. Mechanisms of Immunomodulation by NAG Polysaccharides

### 5.1. Overview

The bioactivities of botanical polysaccharides depend greatly on the chemical structure, the proposed pattern recognition receptor (PRR), and the signal transduction pathway initiated. The potential receptors and mechanism of action of plant polysaccharides has been heavily reviewed by Tzianabos in 2000 [64], Schepetkin and Quinn in 2006 [9], and recently updated by Yin et al. in 2019 [65], however, the receptors and mechanism of action for NAG polysaccharides remain elusive. The structure of plant polysaccharides mediates an important role in PRR specificity, and binding of different PRRs often leads to a specific inflammatory response [9,65]. Though PRR specificity and polysaccharide heterogeneity may provide caveats in the hypotheses regarding which PRR the different NAG polysaccharides bind to, there is room for postulation of potential PRRs for NAG polysaccharides by comparing their structure with other plant polysaccharides whose PRR and signaling pathways have already been determined. In this section, we review the most common receptors of the innate immune system which may potentially bind to NAG polysaccharides, compare various NAG polysaccharides with other polysaccharides of similar structure, and examine which signaling pathway is most likely modulated by those polysaccharides. Most of the signaling pathways described below are illustrated in Figure 3.

### 5.2. Receptors of the Innate Immune System that Potentially Bind to NAG Polysaccharides

#### 5.2.1. TLRs

The Toll-like receptors (TLR) are one of the most heavily studied type of cell surface receptor involved in immunomodulation [66,67,68,69,70,71,72,73]. Though there are 10 different TLRs found in human macrophages and 12 in murine macrophages, polysaccharides seem to mainly be associated with TLR2 and TLR4 binding [65,71]. TLR4 has been shown to be the primary pattern recognition receptor for lipopolysaccharide (LPS; endotoxin) on the surface of Gram-negative bacteria [74,75,76]. TLR4 has also been shown to be reactive to many types of plant polysaccharides [77]. Upon TLR4 activation, the Toll-interleukin 1 receptor (TIR) domain containing the adaptor protein (TIRAP) binds to TLR4 recruiting myeloid differentiation primary response gene 88 (MyD88), which activates interleukin 1 (IL-1) receptor associated kinases (IRAK) [78]. This leads to the activation of transforming growth factor-β-activated kinase 1 (TAK1) through TNF receptor associated factor 6 [76]. TAK1 then activates NF-κB through the inhibitor of κB (IκB) kinase (IKK) complex mediated phosphorylation and ubiquitination of IκBα, and also activates the different mitogen activated protein kinases (MAPKs, JNK, p38, ERK1/2) [76,78,79]. TLR4 can also activate NF-κB independently from MyD88 through TIR-domain-containing adapter-inducing interferon-β (TRIF) [79]. This leads to the endocytosis of TLR4 and eventual activation of receptor-interacting protein 1 (RIP1), which leads to TAK1 activation [67,69,76,80]. TLR2 heterodimers utilize an almost identical mechanism to the TLR4-MyD88 signaling pathway but have differing substrate specificity [67]. Additionally, TLR1 and 6 are known to dimerize with TLR2 allowing for recognition of a large degree of substrates [72]. The activation of these pathways leads to an increase in the production of proinflammatory cytokines, such as TNF-α and various interleukins (IL), reactive nitrogen species (RNS), reactive oxygen species (ROS), prostaglandins such as prostaglandin E_2_ (PGE_2_), and type I interferons, as well as increased cell proliferation. Certain TLRs such as TLR3, TLR7, human TLR8, and TLR9 are found on the endoplasmic reticulum and endosomal compartments, and they bind mainly to single- or double-stranded foreign nucleic acids [72]. Though TLR3 and the other endosomal TLRs bind to similar substrates, their signaling is quite different, whereas TLR3 activates TRIF, the other endosomal TLRs are MyD88 dependent [72]. All TLRs with the exception of TLR3 and TLR4 seem to be MyD88 dependent, and lead to the activation of NF-κB and MAPKs [72]. TLR4 has been indicated as one of the potential receptors for NAG polysaccharides, however, since NAG polysaccharides have been shown to activate the NF-κB, MAPK, and PI3K signal transduction pathways there are potentially other receptors involved [30,37,49].

#### 5.2.2. C-type Lectin Receptors

C-type lectin receptors are endocytic receptors with a high amount of carbohydrate recognition domains and have been shown to be a key receptor in the recognition of multiple polysaccharides [9,65]. These receptors have been postulated to be potential receptors for polysaccharides isolated from both *P. ginseng* and NAG [81]. Of these receptors, the β-glucan receptor dectin-1, has been the most studied. Other c-type lectin receptors include DC-specific intercellular adhesion molecule-3-grabbing non-integrin (DC-SIGN), SIGN-R1, mannose receptor (MR), macrophage inducible c-type lectin (mincle), collectin, and selectin, which bind to a variety of polysaccharides structures [81,82]. With the exception of mannose receptor and DC-SIGN, c-type lectin receptors have been reported to activate spleen tyrosine kinase (Syk), and leads to the activation of protein kinase C-δ (PKCδ), possibly through the activation of phospholipase-Cγ (PLCγ) [82,83,84]. This in turn activates caspase activation and recruitment domain-containing protein 9 (CARD-9), which can interact with B-cell lymphoma/leukemia 10 (BCL-10), mucosa-associated lymphoid tissue lymphoma translocation protein 1 (MALT1), which leads to NF-κB and inflammasome activation, or alternatively it can lead to Ras-GRF-1/H-Ras dependent activation of MAPKs [82,83,84]. Additionally, Syk can also activate the nuclear factor of activated T-cells (NFAT) through the calcium and calmodulin dependent serine-threonine kinase calcineurin, leading to the induction of cyclooxygenase 2 and various proinflammatory interleukins [84,85]. DC-SIGN, as well as dectin-1, has been reported to activate NF-κB through Ras/Raf-1 signaling in a Syk independent manner [84]. Ligation of MR has been reported to lead to phagocytosis through interaction with mannose containing polysaccharides, and possibly to activation of an inflammatory response through interactions with TLR2 but has been also linked to an anti-inflammatory activity [9,65,86,87,88,89].

#### 5.2.3. CR3, SRs, and NOD-2

Other receptors, such as complement receptor 3 (CR3), scavenger receptors (SRs), and nuclear oligomerization domain-2 (NOD-2), have been shown to bind to polysaccharides causing an immunostimulating response [9,65,90,91]. CR3, a member of the beta-integrins [92] is known to cause the activation of Syk, leading to the activation of phosphatidylinositol 3-kinase [93]. The activation of CR3 has been shown to cause increased NF-κB and MAPK activation potentially through AKT in macrophages [94,95,96]. However, pretreating innate immune cells using certain agonists for CR3 has also been shown to reduce inflammation caused by TLR2, 7, and 8 [97,98], which may help explain the modulatory effect of some polysaccharides on stimulated macrophages [99]. SRs have been shown to stimulate MAPK activation through PLCγ and PKC but have also been shown to utilize a Src-Rac1-PAK MAPK (JNK/p38, not ERK) to upregulate TNF-α and IL-1β expression, and activate PI3K depending on the substrate [65,100,101,102]. Different scavenger receptors, such as SCARF1 and CD36, which bind to β-glucans, have also been shown to work in conjunction with TLR2 [103,104,105]. NOD-2 is a member of the NOD—leucine rich repeat (LRR), or NOD-like receptor (NLR) family of receptors, which is known to bind peptidoglycans and can be activated by muramyl dipeptide, but has also been shown to bind partially digested polysaccharides [90,106,107]. NOD-2 then signals to RIP-2 (also known as RICK), which leads to the IKK activation through TAK1 signaling [107,108]. In the following section, we will discuss the structural features of NAG polysaccharides and postulate on their potential receptors and downstream mechanisms based on other plant polysaccharides with a similar structure and composition.

### 5.3. NAG Polysaccharide Classes and Their Immunomodulatory Mechanism

A neutral polysaccharide (AGC1) containing type II arabinogalactans isolated from the NAG cell suspension culture has been shown to cause an increase in the production of proinflammatory cytokines, iNOS, and NO from RAW264.7 murine macrophages [14]. This increase was partially attenuated in the presence of Bay-11, a known inhibitor of IKK, indicating the involvement of NF-κB [14,109]. Additionally, AGC1 was also shown to increase the transcription of *Nos2*, the gene coding for iNOS, as shown by an increase of cytoplasmic mCherry, under control of the *Nos2* promoter [14]. AGC1 was also shown to increase the production of TNF-α and nitrite, as well as increase proliferation in extracted murine splenocytes [14]. Several other arabinogalactans isolated from other plants have also shown promising immunostimulatory activity [110,111,112,113]. A type II arabinogalactan isolated from *Acanthopanax sciadophylloides* was shown to increase levels of proinflammatory cytokine mRNA, increase NO production, and activate NF-κB in RAW264.7 murine macrophages [114]. This was diminished upon inhibition of either TLR4 or a cluster of differentiation factor 14 (CD14), a known coreceptor of TLR4 [114]. Conversely, another type-II arabinogalactan from *Anoectochilus formosanus* (AGAF) caused the activation of NF-κB and MAPK (ERK and p38), which was partially attenuated upon inhibition of Dectin-1 and TLR2, but not TLR4 [115]. These examples provide evidence to suggest either the TLR4-CD14-dependent pathway or Dectin-1/TLR2 signaling pathways as potential mechanisms of action for AGC1 and other type-II arabinogalactans from NAG, which may be fully characterized in the future.

An acidic polysaccharide containing RG-I pectin isolated from the NAG cell suspension culture has been shown to modulate the inflammatory response of RAW 264.7 cells increasing both the activation of NF-κB and p38 (MAPK) [37]. Though there are many different receptors which can lead to the activation of these two signaling pathways, pectins isolated from unripe papayas (*Carica papaya* L; Un-1-WSF/Un-2-WSF), have been shown to induce an inflammatory response in THP-1 monocytes primarily through MyD88-dependent signaling [90]. Un-1-WSF/Un-2-WSF greatly induced the NF-κB release and activation in HEK 293T cells expressing TLR2 and TLR4, but attenuated TLR3 and 9 agonist induction of NF-κB [90]. It is important to note that the ripening-induced methyl esterification of both polysaccharides caused NF-κB activation through TLR2,3,4,5, and 9, as well as NOD-2, but only in large concentrations [90]. Methyl-esterification of pectic polysaccharides has been shown to increase TLR4/MyD88 signaling leading to an increased inflammatory response [116]. Another RG-I isolated from peels of the Korean green citrus fruit, *Cheongkyool* (CCE-I), was shown to activate p65 (NF-κB) nuclear translocation, and activate MAPKs, leading to the production of nitric oxide (NO) and IL-6, in RAW 264.7 macrophages [117]. Using various receptor neutralizing antibodies, the authors determined that CCE-I activated RAW 264.7 macrophages mostly through interactions with SRs [117]. The literature cited implies a possible TLR/SR dependent activation of NF-κB and MAPK (p38) for RG-I polysaccharides isolated from NAG. Though there are few publications regarding the signaling pathway of NAG polysaccharides, by comparing characterized structures, a potential mechanism of action for their immunostimulatory activity may be hypothesized.

Many polysaccharides isolated from NAG are yet to be fully characterized making it difficult to predict the potential mechanism of action. Earlier, we discussed the immunomodulatory activity and monosaccharide composition of crude NAG polysaccharides and their ability to increase RNS and proinflammatory cytokine production in macrophages [12,29,30,31,35,36,38,47,48,49]. Here, we attempt to compare partially characterized NAG polysaccharides to other plant polysaccharides based on the monosaccharide composition, but it should be noted that without further structural analyses it is not possible to infer a potential mechanism of action. A polysaccharide from *Ganoderma atrum* (PSG-1) similar in monosaccharide composition, but not identical to PPQA4, AEP-1, and AEP-2 [31,38], was shown to elicit an immune response in non-stimulated peritoneal macrophages, as well as attenuate an LPS-induced immune response through interactions with MR [86]. The authors hypothesized that PSG-1 binds to both TLR4 and MR, mediating the inflammatory response of peritoneal macrophages to LPS, through an unknown mechanism [86]. A polysaccharide similar in monosaccharide composition to PSG-1 (DPI) was shown to increase the immune response of RAW 264.7 macrophages through CR3 [118]. Additionally, a red ginseng acidic polysaccharide (RGAP) isolated from *Panax ginseng*, similar in monosaccharide composition to PPQA2 and PPQA5 [38] was shown to induce the immune response of RAW 264.7 macrophages, increasing the NO production by activation of ERK and JNK (MAPKs) mostly through the interaction with TLR2 [119,120]. These together show that TLR4, TLR2, MR, and CR3 may be potential receptors for these partially characterized NAG polysaccharides. However, as previously implied, though the monosaccharide composition of polysaccharides may be similar, the immunomodulatory activity, potential receptors, and subsequent mechanism of action for those polysaccharides may differ greatly. For example, a glucogalactan from NAG (PPQ) was shown to cause an increase in IL-2 and IFN-γ expression, as well as an increase in the size of the thymus and spleen in male C57BL/6 mice, though another similar NAG glucogalactan, PPQN, showed an anti-inflammatory activity rather than immunostimulation [47,48]. Additionally, PSG-1 and DPI have similar monosaccharide compositions, but have quite different proposed mechanisms of action [86,118]. These differences may be due to the glycosyl linkages, indicating that the complete characterization of crude NAG polysaccharides is essential for determining any structural activity relationship. Additionally, mild acid hydrolysis or enzymatic digestion of polysaccharides may be a useful strategy to elucidate structure-activity relationships. The hydrolyzed oligosaccharides can be used to further evaluate the immunomodulatory activity and identify specific bioactive glycan residues.

Though many NAG polysaccharides have shown to be immunomodulatory, the mechanism of activation, and the receptors involved, remain elusive. It follows that polysaccharides from other plants, with similar monosaccharide compositions and structures as those from NAG, can be used to formulate a speculative hypothesis. In this section, we have outlined the various pathways for known polysaccharide receptors and reviewed mechanisms for different NAG polysaccharides. We also compared NAG polysaccharides to polysaccharides with a similar composition/structure, whose receptor and mechanism of action had already been determined. In doing so, three distinct trends were identified that may influence future research regarding NAG polysaccharides. First, most polysaccharides, similar in structure to NAG polysaccharides, interacted primarily with either TLR4, TLR2, SR, or Dectin-1, indicating their potential as NAG polysaccharide receptors. Second, most macrophage receptors which can bind to polysaccharides, with the exception of DC-SIGN, MR, and Dectin-1, initially induce a somewhat distinct signaling cascade, which converges with the activation of NF-κB (p65) and/or MAPK (p38/ERK/JNK), and have a significant overlap/cross-talk with one another. This indicates a need for more thorough investigation of the PRR/mechanism of action involved in the NAG polysaccharide immunomodulatory response. Finally, polysaccharides which have similar monosaccharide compositions may exhibit different immunomodulatory activities, bind to different receptors, and activate different pathways. These complex cellular responses need further characterization of NAG polysaccharides to determine a potential structure-activity relationship. Overall, the interaction of mammalian cells and the polysaccharides of NAG continues to be of interest and determination of a potential structural activity relationship will be important.

## 6. Recent Innovations and Future Directions

Despite progress in NAG polysaccharide research over the last two decades, significant challenges still remain. Apart from ongoing research focusing on the elucidation of a structure-activity relationship, recent innovations have addressed other existing challenges in NAG polysaccharide research such as standardization of polysaccharide extracts and endotoxin contamination. The batch-to-batch structural consistency of polysaccharides is critical towards the application of these compounds as nutraceuticals or therapeutics. Additionally, the accurate measurement of endotoxin levels in polysaccharide samples is also crucial for the safety and eventual efficacy of these compounds. NAG roots are often contaminated with high levels of endotoxin owing to Gram negative bacteria in the soil. An endotoxin clean-up step using endotoxin removal columns is usually necessary prior to the biological assays. Endotoxin levels as low as 500 pg/µg in polysaccharide samples can elicit a substantial immunostimulatory response leading to false positives. The lack of clean-up procedures and endotoxin testing can lead to misleading claims about the immunomodulatory effects of polysaccharides. Although this issue has been highlighted by several researchers, a large number of recent studies do not report endotoxin levels in their extracts or purified fractions. A greater emphasis towards reporting endotoxin levels in polysaccharide samples will establish the credibility of these compounds as immunomodulators. Recently, it has been shown that the NAG tissue culture can be used as a viable platform for the production of polysaccharides with consistent structural characteristics and reduced endotoxin content [14,37]. Historically, the tissue culture has been widely used for the production of valuable secondary metabolites, but the technique was not extensively used for the production of primary metabolites such as polysaccharides. It is well known that the tissue culture can produce similar compounds that whole plants produce in a controlled environment without regard to the season, environmental conditions, or climate change [121,122]. The cultured cells are generally pathogen-free, making downstream processing easier and potentially inexpensive. The platform can potentially eliminate the batch-to-batch structural inconsistency as metabolites are produced in a controlled environment in a definite amount of time. The platform is not only suitable for large-scale production of standardized polysaccharides but also promotes conservation efforts. Although the crude polysaccharide yield using this platform was found to be lower than dried NAG roots, research is ongoing to increase yields for potential commercial applications. Thus far, a number of immunomodulatory polysaccharides have been purified and partially characterized from NAG suspension cultures and these compounds can be potentially used as nutraceuticals in the future [14,37].

The quality control of polysaccharide-based nutraceuticals is often time consuming and challenging due to the lack of a single analytical platform capable of analyzing the purity, molecular weight of constituent polysaccharides, and monosaccharide composition. Laborious derivatization techniques for the monosaccharide composition analysis was a major bottleneck in the pipeline for rapid quality control of commercial polysaccharides. Recently, the direct analysis of monosaccharide composition without derivatization has accelerated the quality control process. The HPAEC-PAD and HPLC-ELSD/CAD methods for separation and detection of neutral and acidic monosaccharides are becoming increasingly popular for the analysis of polysaccharides [40,41,42,43]. Semi-universal detectors such as CAD are suitable for the detection of non-UV and weakly-UV active compounds. Detectors such as CAD are inexpensive and easy to use having a few controllable parameters compared to a mass spectrometer. HPLC coupled to CAD has been shown to be effective as a multipurpose quality control analytical tool for (i) underivatized monosaccharide composition analysis and (ii) determination of purity, homogeneity, and molecular weight of polysaccharides [42]. These techniques will provide a robust platform for rapid and accurate quality control of NAG polysaccharides.

Although, NAG polysaccharides are largely effective in eliciting an immune response in cell-based assays, the in vivo pharmacological effects may be limited due to the polysaccharide’s large molecular size, heterogeneity, and poor solubility [52,54]. Recently, NAG polysaccharide nanoparticles were prepared (average size of 20 ± 4 nm) and encapsulated within biodegradable gelatin nanospheres to enhance their oral drug delivery [54]. This approach of oral drug delivery was shown to enhance the immunomodulatory properties of NAG polysaccharides in Swiss albino mice. Such approaches will be crucial towards the formulation of truly effective NAG polysaccharide-based therapeutics.

NAG polysaccharide research is also expected to benefit from advancements in metabolomics analyses. Metabolomics can accurately depict the components present in NAG aqueous extracts including sugars and other impurities. Not only can this approach help in standardization efforts but also predict the global impact of these extracts on primary metabolism. The body fluids (e.g., blood and urine) collected from individuals who consume NAG polysaccharide nutraceuticals can be analyzed by an untargeted metabolomics approach to potentially identify biochemical mediators affected by these polysaccharides. Additionally, the same supernatants used for cytokine analysis during in vitro cell-based assays can be used for an untargeted metabolomics analysis to elucidate the global effect of these compounds on human metabolism. We expect a significant use of metabolomics to elucidate the bioactivities of plant polysaccharides including NAG in the upcoming years.

## 7. Conclusions

Research on NAG polysaccharides over the last two decades has resulted in the extraction and purification of multiple immunomodulatory compounds. Some of these compounds are currently sold as nutraceuticals and dietary supplements. Several other NAG polysaccharides have the potential to be formulated into therapeutics in the future due to their potent immunomodulatory effects. However, the mechanisms of action of these compounds are unclear. Due to the lack of complete structural information of NAG polysaccharides, it is also difficult to predict structure-activity relationships. Nevertheless, this review summarized the structural features of crude, partially purified and pure NAG polysaccharides and attempted to correlate the chemical composition to immunomodulatory properties. The potential receptors of the innate immune system that bind to these compounds and the putative mechanisms of action were extensively discussed. Due to its worldwide popularity, research in NAG polysaccharides is expected to continue to expand. As techniques for carbohydrate characterization improve and become more accessible, we expect NAG polysaccharides to be completely characterized along with the elucidation of structure-function relationships and modes of action. Major challenges such as the standardization of NAG polysaccharide-based nutraceuticals, endotoxin contamination, and proper quality control of extracts will need to be further addressed. Innovations such as the application of plant tissue culture to produce natural polysaccharides can be promising for commercial purposes. Additionally, the use of metabolomics in NAG polysaccharide research has the potential to drive further progress in the field.

## Figures and Tables

**Figure 1 molecules-25-05854-f001:**
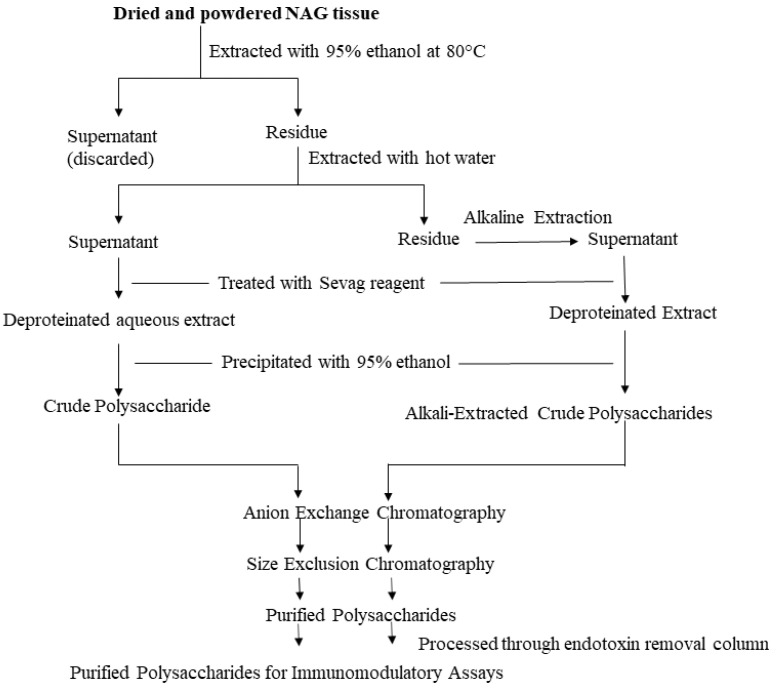
Extraction and Purification Workflow. The illustrated workflow is recommended for the purification of polysaccharides intended for immunomodulatory assays.

**Figure 2 molecules-25-05854-f002:**
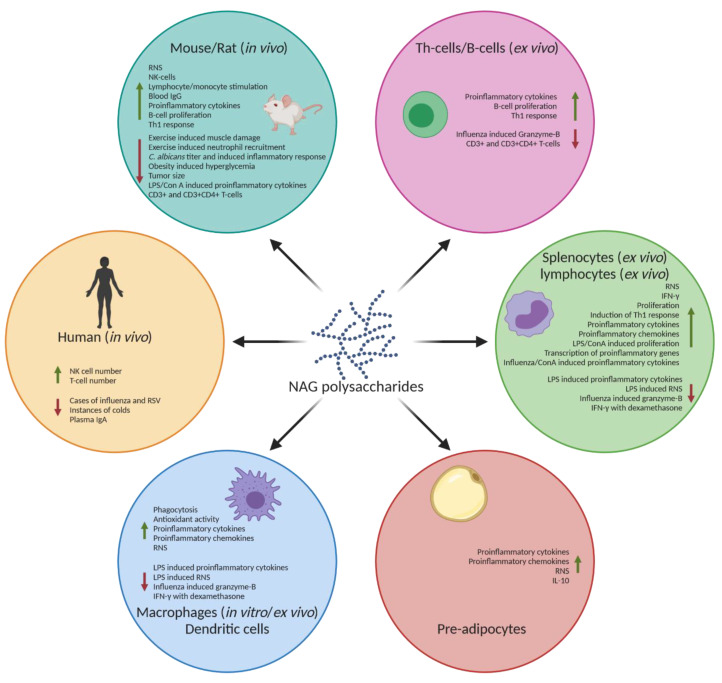
Immunomodulatory effects of NAG polysaccharides. NAG polysaccharides have been shown to cause an immunostimulatory response, such as increased RNS, IFN-γ, and proinflammatory cytokines, such as TNF-α and various interleukins in various in vitro and in vivo models. NAG polysaccharides have also been shown to reduce these proinflammatory products in already activated immune systems, as well as increase the production of anti-inflammatory cytokines such as IL-10. The green arrows indicate an increase. The red arrows indicate a decrease. Created with BioRender.com.

**Figure 3 molecules-25-05854-f003:**
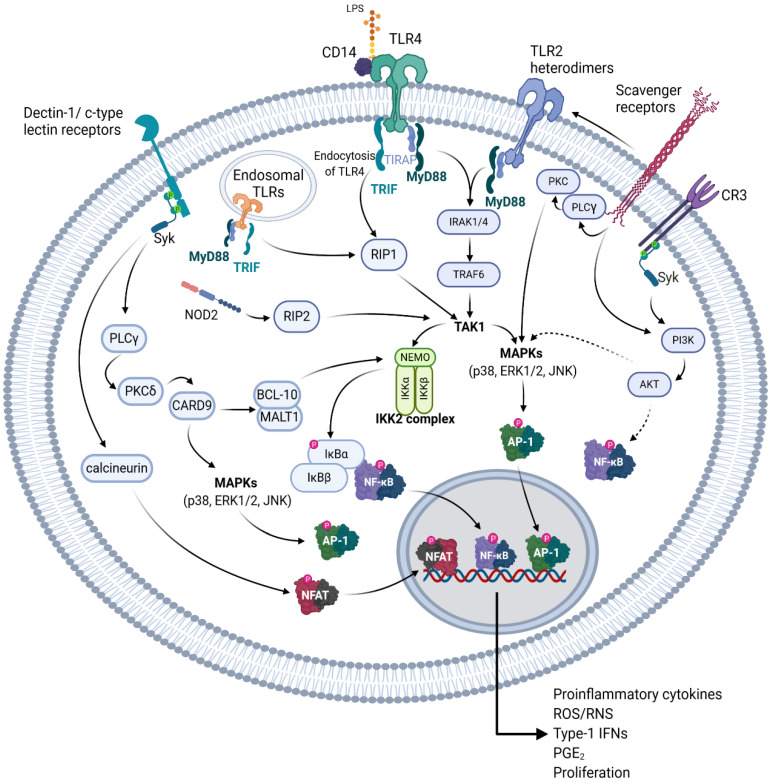
Receptors that potentially bind to NAG polysaccharides and the respective signal transduction pathways initiated. A graphical representation depicting various pattern recognition receptors (PRR) of the innate immune system, which have been shown to recognize various structural components of NAG polysaccharides, or similar polysaccharides of other plants. Solid arrows indicate a probable direct activation, dotted arrows indicate unknown intermediate steps. These pathways are detailed in Section 5.1. Created with BioRender.com.

**Table 1 molecules-25-05854-t001:** Structural features of polysaccharides isolated from different sources of North American ginseng (NAG).

Name	Structural Features of Isolated Polysaccharides	Molecular Weight (kDa)	Ref
AGC1	Composed of Gal (60.093%), Ara (19.165%), Xyl (11.363%), Glc (6.298%), Rha (1.548%), and Man (0.79%); contains type II arabinogalactans	5.2	[14]
AGC3	Composed of Ara (7.8%), Rha (8.1%), Glc 92%), Gal (74.3%), GalA (6.8%), GluA (1%), and trace amounts of Man and Xyl; contains RG-I polysaccharide	4.81 and 32.14	[37]
GSP	Composed of Rha, Ara, Gal, Glc, and GalA in a molar ratio of 1:4:8:8:50; pectic polysaccharide with a major homogalacturonan domain and a minor RG-I component	85.4	[33]
PPQN	Composed of Glc and Gal in a molar ratio of 1:1.15	3.1	[48]
PPQA2	Composed of Ara, Rha, Man, Gal, Glc, GalA, and GluA in a molar ratio of 8:4:2.9:7.2:12.5:26.6:38.8	23	[38]
PPQA4	Composed of Ara, Rha, Man, Gal, Glc, and GluA in a molar ratio of 19.7:5.1:8.1:23.9:41.3:2	120	[38]
PPQA5	Composed of Ara, Rha, Man, Gal, Glc, GalA, and GluA in a molar ratio of 8.5:3.2:5.3:10.8:32.4:15.5:24.4	5.3	[38]
AEP-1	Composed of Glc, Gal, and GalA in a molar ratio of 4.67:0.97:3.92	N/A	[31]
AEP-2	Composed of Ara, Man, Gal, Glc, and GalA in a relative molar ratio of 1.03:0.76:1.68:3.02:3.65	N/A	[31]
WPS-1	Composed of Ara, Rha, Man, Gal, and Glc in a ratio of 21.2:2.3:2.6:18.7:55.2	1540	[35]
WPS-2	Composed of Ara, Rha, Man, Gal, and Glc in a ratio of 27.9:1.7:2.9:20.7:46.8	14.1	[35]
SPS-1	Composed of Ara, Xyl, Man, Gal, Glc, Gala, and GlcA in a ratio of 22.3:6.9:9.2:28.6:15.9:13.6:3.5	362	[35]
SPS-2	Composed of Ara, Xyl, Man, Gal, Glc, Gala, and GlcA in a ratio of 14.2:5.3:7.9:22.5:25.3:16.9:7.9	9700	[35]
SPS-3	Composed of Ara, Rha, Xyl, Man, Gal, Glc, GalA, and GlcA in a ratio of 19.2:2.1:9.6:12:15.2:11.5:26.3:4.1	512	[35]
PPQ	Composed of Glc and Gal in a molar ratio of 2.1:1	54	[47]
Quinquefolan A	Composed of Man and Glc in a ratio of 1.0:2.3 and 10.8% uronic acid	>2000	[46]
Quinquefolan B	Composed of Man and Glc in a ratio of 1.0:5.5 and 11.7% uronic acid	>2000	[46]
Quinquefolan C	Composed of Xyl and 7.1% uronic acid	>2000	[46]

**Table 2 molecules-25-05854-t002:** Immunomodulatory activities of NAG crude and purified polysaccharides.

Name of Polysaccharide	Immunomodulatory Findings	Model	Endotoxin Test	Ref
N/A (Crude polysaccharide)	Stimulated TNF-α	Alveolar macrophages isolated from male Wistar rats	Yes	[29]
N/A (Polysaccharide Nanoparticles)	Stimulated NO, TNF-α, IL-6, and IL-1β	RAW 264.7 murine macrophage cells	Not reported	[52]
N/A (Fluorescein-5-thiosemicarbazide labelled polysaccharide nanoparticles	Uptake of nanoparticles by macrophages, stimulated NO, TNF-α, IL-6, and IL-1β	RAW 264.7 murine macrophage cells	Not reported	[53]
N/A Gelatin-encapsulated Polysaccharide Nanoparticles	Stimulated NO, TNF-α, IL-6, and IL-1B (in vitro); stimulated NO and TNF-α in blood serum (in vivo)	RAW 264.7 murine macrophage cells (in vitro); swiss albino mice (in vivo)	Not reported	[54]
N/A (crude and partially purified polysaccharide fractions of different molecular weights)	Stimulated NO and TNF-α (in vitro, ex vivo, and in vivo); suppressed NO and TNF-α in LPS-stimulated cells (in vitro, ex vivo, and in vivo)	RAW 264.7 murine macrophage cells (in vitro); alveolar macrophages isolated from adult male rats (ex vivo); adult male rats (in vivo)	Not reported	[12]
AGC1 (Purified neutral polysaccharide)	Stimulated TNF-α, IL-6, MCP-1, GM-CSF, NOS2 gene expression, iNOS protein expression, NO (in vitro); stimulated splenocyte proliferation (ex vivo)	RAW 264.7 murine macrophage cells (in vitro); splenocytes isolated from CD1 mice (ex vivo)	Yes	[14]
AGC3 (Purified acidic polysaccharide fraction)	Stimulated TNF-α, IL-6, and NO (in vitro); upregulated phospho-p65 and phospho-p38 (in vitro); stimulated splenocyte proliferation (ex vivo)	RAW 264.7 murine macrophage cells (in vitro); splenocytes isolated from CD1 mice (ex vivo)	Yes	[37]
N/A (High molecular weight crude polysaccharides)	Stimulated IL-6, TNF-α, IFN-y, IL-1β, IL-12, IL-2; induction of Th1 immune response; induced NF-κB, MAPK and PI3K pathways	Human peripheral blood mononuclear cells, CD14+ monocytes, CD14+CD16+ monocytes	Yes	[30]
PPQN (Purified neutral polysaccharide)	Suppressed NO, IL-1β, IL-6, and TNF-α in LPS-stimulated macrophages	RAW 264.7 murine macrophage cells	Not reported	[48]
PPQA2, PPQA4, PPQA5 (Purified acidic polysaccharides)	Stimulated NO, TNF-α, and IL-6	RAW 264.7 murine macrophage cells	Not reported	[38]
N/A (Crude polysaccharide)	Stimulated IL-6, CCL5, TNF-α, and NF-KB	Mouse 3T3-L1 preadipocyte cells	Not reported	[49]
AGP (Crude Polysaccharide)	Stimulated NO, IL-6, and IL-10	RAW 264.7 murine macrophage cells	Not reported	[34]
AEP (alkali-extractable crude polysaccharide), AEP-2 (Purified acidic polysaccharide)	Stimulated NO, IL-6, and TNF-α	RAW 264.7 murine macrophage cells	Not reported	[31]
CPS (crude polysaccharide), WPS-1, WPS-2 (purified neutral polysaccharides), SPS-1, SPS-2, and SPS-3 (purified acidic polysaccharides)	Stimulated macrophage phagocytic activity and NO; augmented LPS and ConA induced splenocytic proliferation (SPS-3 > SPS-1 > CPS> WPS-1 > WPS-2 > SPS-2)	Splenocytes and peritoneal macrophages isolated from specific pathogen-free Jilin mice	Not reported	[35]
PPQ (purified neutral polysaccharide)	Decreased tumor size; stimulated serum IL-2 and IFN-γ; reduced serum IL-10; increased thymus and spleen index in tumor-bearing mice	C57BL/6 mice (Lewis lung carcinoma model)	Not reported	[47]

**Table 3 molecules-25-05854-t003:** Immunomodulatory activities of CVT-E002/COLD-FX.

Immunomodulatory Activities of CVT-E002/COLD-FX	Model	Ref
Decreased spleen IL-2 and IFN-γ and increased IL-1β following LPS/ConA stimulation of cultured lymphocytes; lowered proportion of CD3+ (T-cells) and activated T-cells	Sprague-Dawley rats (oral administration) followed by isolation and culture of lymphocytes from spleen, mesenteric lymph nodes, and Peyer’s patches	[56]
Reduced acute respiratory illness due to influenza or respiratory syncytial virus	Randomized, placebo controlled, double blind human clinical trial (oral administration)	[57]
Stimulated TNF-α and IL-2; moderated granzyme-B levels	Human (oral administration) followed by isolation and culture of peripheral blood leukocytes along with influenza viruses	[58]
Increased NK cell numbers in spleen, bone marrow, and blood	C3H/OuJ mice (oral administration)	[59]
Reduced recurrent and mean number of colds per person and severity of symptoms	Randomized, placebo controlled, double blind human trial (oral administration)	[60]
Increased NK cell and T-helper cell numbers and reduced IgA levels in plasma	Randomized, placebo controlled, double blind human trial (oral administration)	[61]
Stimulated B lymphocyte proliferation; stimulated IgG in blood serum; stimulated IL-1, IL-6, TNF-α, and NO	Lymphocytes isolated from BALB/c mice; BALB/c mice (oral administration); peritoneal exudate macrophages isolated from C57BL/6 mice	[45]
Stimulated ConA-induced IFN-γ and IL-2	Splenocytes isolated from C57BL/6 mice	[55]
Suppressed IFN-γ and related gene expression in dexamethasone cotreated cells	Mouse splenocytes	[62]
Inhibited allergic airway inflammation; increased INF-γ, regulatory T-cells, and IL-10 in lungs	BALB/c mice	[63]

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
