# Peer review of "Panax quinquefolius (North American Ginseng) Polysaccharides as Immunomodulators: Current Research Status and Future Directions"

_molecules, 2020, doi:10.3390/molecules25245854_

Round 1
Reviewer 1 Report
This manuscript presents the review for the structural features and the immunomodulatory properties of polysaccharides from Panax quinquefolius. The manuscript seems to be carefully and clearly presented.
Author Response
Response to Reviewer 1 Comments
Point 1: “This manuscript presents the review for the structural features and the immunomodulatory properties of polysaccharides from Panax quinquefolius. The manuscript seems to be carefully and clearly presented.”
Response: Thank you for reviewing our submission.
Reviewer 2 Report
The review manuscript has a high quality, I always do the reviews in a conciseand impartial manner. I checked several points in manuscript and authors
addressed more than expected for theme, for example the immune activation
mechanism for each NAG polysaccharide. In addition, the English language is
refined with rare mistakes of some words. The only question would be in reference to figure 1, quality can be improved. Again I recommend that you accept the manuscript. The quality and originality
of theme, will generate several citations contributing to your valuable journal.
Author Response
Response to Reviewer 2 Comments
Point 1: “The review manuscript has a high quality, I always do the reviews in a concise and impartial manner. I checked several points in manuscript and authors
addressed more than expected for theme, for example the immune activation
mechanism for each NAG polysaccharide. In addition, the English language is
refined with rare mistakes of some words. The only question would be in reference to figure 1, quality can be improved. Again I recommend that you accept the manuscript. The quality and originality of theme, will generate several citations contributing to your valuable journal.”
Response 1: Thank you for the kind words! The quality of figure 1 has been improved.
Reviewer 3 Report
- In lines 82 and 83 the reference 24 is cited, it is suggested to rewrite to put the reference only once.
- In line 136 the citations are in subscript; format must be corrected.
- Check that the format of the citations is the same throughout the document, for example in line 21, the cite has a comma (et al.,) and in line 213 it does not have a comma (et al.).
- Avoid repeating information in the text and tables, for example, the information of at least 10 NAG polysaccharides is repeated in the text and the table, review the entire document and all the tables.
- Verify that the font size is the same throughout the manuscript, for example, there are different font sizes in lines 353-370.
- Verify that the figures and tables are cited in the text before they appear, for example figure 3 is on page 12 and the figure was cited on page 13.
- The authors indicate that an important factor is the presence of endotoxins as a contaminant of NGA polysaccharides that act as immunomodulators. It is suggested that information be included in this regard, such as type, concentration and origin of endotoxins.
- Verify that all references have the same format and according to the instructions for author, for example, in some, the name of the journal is abbreviated, in some the title is in bold.
